# Hybrid Approach for Wood Modification: Characterization and Evaluation of Weathering Resistance of Coatings on Acetylated Wood

Anna Sandak [1,2], Edit Földvári-Nagy [1], Faksawat Poohphajai [1,3], Rene Herrera Diaz [1,4], Oihana Gordobil [1], Nežka Sajinčič [1], Veerapandian Ponnuchamy [1,5] and Jakub Sandak [1,5,*]

1 InnoRenew CoE, Livade 6, 6310 Izola, Slovenia; anna.sandak@innorenew.eu (A.S.); edit.foldvari-nagy@innorenew.eu (E.F.-N.); faksawat.poohphajai@innorenew.eu (F.P.); rene.herdiaz@innorenew.eu (R.H.D.); oihana.gordobil@innorenew.eu (O.G.); nezka.sajincic@innorenew.eu (N.S.); veerapandian.ponnuchamy@innorenew.eu (V.P.)
2 Faculty of Mathematics, Natural Sciences and Information Technologies, University of Primorska, Glagoljaška 8, 6000 Koper, Slovenia
3 Department of Bioproducts and Biosystems, School of Chemical Engineering, Aalto University, P.O. Box 16300, 00076 Aalto, Finland
4 Department of Chemical and Environmental Engineering, University of the Basque Country, Plaza Europa, 1, 20018 Donostia-San Sebastian, Spain
5 Andrej Marušič Institute, University of Primorska, Titov trg 4, 6000 Koper, Slovenia
* Correspondence: jakub.sandak@innorenew.eu; Tel.: +386-40282959

**Abstract:** Wood, as a biological material, is sensitive to environmental conditions and microorganisms; therefore, wood products require protective measures to extend their service life in outdoor applications. Several modification processes are available for the improvement of wood properties, including commercially available solutions. Among the chemical treatments, acetylation by acetic anhydride is one of the most effective methods to induce chemical changes in the constitutive polymers at the cellular wall level. Acetylation reduces wood shrinkage-swelling, increases its durability against biotic agents, improves UV resistance and reduces surface erosion. However, even if the expected service life for external cladding of acetylated wood is estimated to be 60 years, the aesthetics change rapidly during the first years of exposure. Hybrid, or fusion, modification includes processes where the positive effect of a single treatment can be multiplied by merging with additional follow-up modifications. This report presents results of the performance tests of wood samples that, besides the modification by means of acetylation, were additionally protected with seven commercially available coatings. Natural weathering was conducted in Northern Italy for 15 months. Samples were characterized with numerous instruments by measuring samples collected from the stand every three months. Superior performance was observed on samples that merged both treatments. It is due to the combined effect of the wood acetylation and surface coating. Limited shrinkage/swelling of the bulk substrate due to chemical treatment substantially reduced stresses of the coating film. Hybrid process, compared to sole acetylation of wood, assured superior visual performance of the wood surface by preserving its original appearance.

**Keywords:** wood modification; natural weathering; acetylation; coatings; service life performance; aesthetic

## 1. Introduction

Biodegradability, identified as an important advantage of biomaterials from a sustainability perspective, is its biggest weakness when considering materials durability. It is especially relevant to all wooden elements or timber structures that are exposed outdoor to environmental factors like humidity/temperature variation, wetting by rain or UV radiation [1]. The selection of naturally resistant species or improvement of the less

durable by means of modification processes is a common solution to minimize degradation of biomaterials in service [2]. However, all actions related to the chemical treatment of biomaterials should be carefully implemented by balancing the benefits with costs and other possible negative consequences. An optional solution is to program a sequence of maintenance actions where façade elements may undertake cleaning, painting or replacement operations. In the majority of cases, wood weathering is an uneven process resulting in diverse deterioration kinetics in different zones on the building [3]. The surface protected by the roof or other architectural detail changes slower than the fully exposed. The same trend can be associated with the exposure side, with cardinal directions resulting in diverse doses of deteriorating factors [4–6]. Wood, as well as other hygroscopic materials, is subject to dimensional distortions caused by changes in the moisture content, air humidity and temperature. Even if a properly designed structure may not be affected by the moisture-induced shrinkage, the dimensional changes may affect the surface coatings creating excessive stresses followed by surface cracks [7,8]. It alters the protection function of the coating and may promote even more excessive decay through the moisture bridges. It is essential to implement best practices of the "protection by design" paradigm to project and construct façade details that allow or compensate for dimensional distortions and minimize related treats [9].

The most critical limitation of using non-protected wood on the building façade is its aesthetical instability, evidenced as appearance changes over time [10]. All materials used on the façade change their outlook during service, but alterations of the majority of bio-based materials are usually more pronounced and occur more rapidly compared to ones that are not bio-based. Under certain circumstances, this may be considered as an advantage, especially when it is a part of the design strategy or intended rapid homogenization with the local context [11]. However, in the majority of cases, the visual changes on the wooden façade claddings are perceived negatively [12]. Unprotected wood exposed to UV rays ultimately changes its colour turning toward grey tonality. In some cases, it may crack or splits at the ends, especially when the façade is not properly designed or boards are wrongly installed. There are several factors affecting weathering intensity of biomaterials surface, such as location, exposure direction, specific microclimate, architectural details, orientation/position of the assembly, use condition or material intrinsic properties [10]. The biotic attack changes surface colour resulting in green or grey-blue spots on the surface of the material [13,14]. Glossy and shiny appearance may turn over time into the matte surface. After a sufficiently long period of moulds' growth, the component may require replacement due to its aesthetical appearance, even though it can still fulfil other functions such as load-bearing capacity, protecting the building envelope from weathering or providing shelter [15].

Modification processes lead to the enhancement of selected wood properties by means of chemical, biological or physical agents. Several alternative treatments, including both active and passive modifications, are recently available [16,17]. The active modification changes the chemical nature of material through chemical, thermo-hygro-mechanical or enzymatic treatments. Conversely, the passive modification does not alter material chemical composition but rather deposits selected functional molecules by means of bulk impregnation or surface treatments [16]. Consequently, various properties of wood are changed to a different extent depending on the modification process and its intensity.

Chemical modification is the result of a chemical agent reaction with wood chemical components resulting in the formation of covalent bonds. Acetylation is the most well-established treatment where acetic anhydride reacts with hydroxyl groups of the cell wall polymers by forming ester bonds. Acetylation slightly improves UV resistance and reduces surface erosion by 50%, which is important while using wood as façade material [18]. The mechanical strength properties of acetylated wood are not considerably different than not-treated wood; however, its durability is substantially improved [19]. Besides bulk modification processes, several techniques affect properties of the surface only without interfering with the interior of the piece [20,21]. The changes of the surface functionalities

affected by the exterior treatments include UV stabilization (e.g., surface esterification), an increase of hydrophobicity (e.g., reaction with silicone polymers) or improvement of the adhesion (e.g., enzymatic treatment, plasma discharge) [22–24]. In addition, alternative processes can be applied with the purpose to improve biomaterial's surface resistance against biotic and abiotic factors. This includes surface coating [8,20] as well as other treatments such as surface densification [25] or surface carbonization (e.g., Shou Sugi Ban) [26]. Surface finishing by diverse coatings, waxes, oils or stains is the most common procedure of the surface treatment, which is highly influencing its service life performance [27]. The resistance of the surface against deterioration in service highly depends on the finishing product quality (chemical formulation), surface preparation (oxidation stage, roughness, wettability, surface free energy) and the application procedure (industrial coating, immersing, brush or spray) [28]. A large variety of commercially available products for surface finishing allow obtaining different appeal, including colour variation, transparency or gloss. The proper use of the surface finishing technologies may contribute significantly to the aesthetical attractiveness of the structure as well as the appearance changes along the service life of the façade [21]. The cost of the finish, including proper surface pre- and post-treatment, may be considerable. However, in a majority of cases, it is economically viable to increase the initial cost of the façade by increasing the thickness of the coating layer, as it may significantly increase the time of maintenance-free use. It has been reported that the change of the coating layer from 30 μm to 50 μm increases the time of the surface resistance for cracking, and the service life period, by a factor of 1.2 [29].

The bulk and surface modification processes may affect one or more functionalities of the biomaterial used for building façade. Even if it improves certain material assets, the positive effect can be multiplied by merging two or more modification processes. Such an approach turns out to be a "hybrid process" and becomes an optimal solution frequently implemented by bio-materials producers [10]. An example of a successful hybrid modification is the surface coating of acetylated or thermally treated wood. The combined effect of the reduced shrinkage/swelling of the bulk substrate and water protecting coating significantly reduces stresses of the coating film preventing it from cracking. Consequently, the surface façade remains intact for a longer period, preserving its original attractive appearance. Benefits obtained by merging different materials and treatments are highly useful to address design limitations and biomaterials' deficiencies. If properly implemented, hybrid modifications can contribute to reducing the environmental burden and economic cost of the façade. It has to be mentioned, however, that some of the modification processes cannot be merged or may affect undesired changes of other material properties. An example may be an increase of the biomaterial brittleness after some modifications that affect its machinability or paint-ability [17]. For that reason, special attention should be directed toward selecting appropriate treatment combinations and extensive quality control of the hybrid modification processes. The goal of this research was to evaluate the combined effect of wood acetylation followed by seven industrial coatings on wood performance when exposed to natural weathering.

## 2. Materials and Methods

### 2.1. Experimental Samples

Acetylated wooden boards manufactured from radiata pine (*Pinus radiata* D. Don), were used for the preparation of experimental samples. The acetylation and coating application was performed in industrial environments by the collaborating companies. One hundred forty–four small blocks ($150 \times 75 \times 20$ mm$^3$, length × width × thickness, respectively) were cut out and randomly separated into two sets. The first set included eighteen samples that were not surface finished and used for the following tests without additional treatments. The second set contained hundred twenty–six samples that were subjected to the supplementary surface coating procedure. It was implemented as a part of the industrial process by applying selected coatings usually used for building façades. Seven alternative coating products manufactured by three producers were tested and

confronted; 144 (= 8 × 6 × 3) samples were analysed in total, including 8 sample types (#A, #B, #C, #D, #E, #F, #G, #H), 6 natural weathering scenarios (0, 3, 6, 9, 12 and 15 months) and 3 replicas for each testing scenario. The summary of technical details regarding experimental samples is presented in Table 1.

**Table 1.** Summary of acetylated Radiata pine samples tested within natural weathering experiment.

| Sample Code | Surface Coating | Appearance |
| --- | --- | --- |
| A | no treatment | natural wood |
| B | hybrid hydro-oil (producer #1) | transparent brown, mat |
| C | preserving stain emulsified with hydro-oil (producer #1) | transparent brown, glossy |
| D | water-based (producer #1) | white, opaque |
| E | waterborne high-gloss (producer #2) | transparent yellow, glossy |
| F | water-based VV MAT (producer #2) | semi-opaque yellow, glossy |
| G | water-based (producer #3) | white, opaque |
| H | water-based impregnating agent (producer #3) | transparent brown, glossy |

*2.2. Weathering Tests*

Natural weathering tests were performed in San Michele, Italy (46°11′15″ N, 11°08′00″ E). The objective of this test was to collect a set of reference data regarding material performance at the different exposure time. Samples were exposed on vertical stands representing a building façade. The stand was oriented to face the southern direction. The weathering experiment was carried out for a total period of 15 months, starting in March 2017. Three replicas were collected from the stand each third month to stop the deterioration progress. Consequently, a collection of samples exposed to 0, 3, 6, 9, 12 and 15 months was gathered. Each exposure period corresponds to a single experimental scenario, and three replica samples assured insight on the reliability and repeatability of results. All samples were stored in a climatic chamber (20 °C, 65% RH) to stabilize conditions before follow-up measurements.

*2.3. Characterization Methods*

2.3.1. Digitalization and Colour Measurement

Samples after conditioning were scanned with an office scanner HP Scanjet 2710 (300 dpi, 24 bit) and saved as TIF files. Colour changes were assessed by means of a spectrometer following the *CIE Lab* system where colour is expressed with three parameters: $L^*$ (lightness), $a^*$ (red-green tone) and $b^*$ (yellow-blue tone). *CIE L\*a\*b\** colours were measured using a MicroFlash 200D spectrophotometer (DataColor Int, Lawrenceville, Illinois, USA). The selected illuminant was D65 and the viewer angle was 10°. All specimens were measured on five randomly selected spots over the weathered surface. The mean values were considered as a representative colour, even if the maximum and minimum readings were preserved to assess the natural variation of the colour distribution.

2.3.2. Gloss

The mode of light reflection from the surfaces was measured using a REFO 60 (Dr. Lange, Düsseldorf, Germany) gloss meter with incidence and reflectance angles of 60°. Ten measurements along and across the fibre direction were taken on each specimen to address the optical heterogeneity of the light reflectance from the wood surface.

2.3.3. Microscopic Observation and 3D Roughness Measurement

Keyence VHX-6000 digital microscope (Keyence, Osaka, Japan) was used for microscopic observation, high magnification image acquisition and 3D surface topography scanning. Colour images were collected with an optical configuration corresponding to ×30 and ×200 magnifications. The light direction and intensity were adjusted to assure a wide dynamic range of the image and avoid saturated pixels generation. Part of the

high magnification images was acquired in the real-time 3D depth reconstruction mode. It allowed post-processing of data to determine surface profile as well as surface roughness indicators. For that reason, an area of $2 \times 2$ mm$^2$ was assessed while series of 3D images stacked together. The proprietary software of the Keyence microscope was used for roughness data post-processing. The protocol included removal of the error of form (plane extraction) and filtering of the surface topography data with a Gaussian band pass filter ($2 \mu m < \lambda < 0.8$ mm). The surface roughness quantifiers determined on such prepared data included arithmetical mean height (*Sa*), skewness (*Ssk*) and kurtosis (*Skt*).

### 2.3.4. Contact Angle and Surface Free Energy

Dynamic contact angle measurements were performed using optical tensiometer Attention Theta Flex Auto 4 (Biolin Scientific, Gothenburg, Sweden). Five replica measurements were performed implementing the sessile drop method on each specimen with a sequence of distilled water and formamide droplets. The dosing volume of each drop was 4 μL controlled by both precise dispenser and drop image analysis. The measurement of the drop shape started at the moment of the initial drop contact with the assessed sample surface and lasted for 20 s. The series of images collected were post-processed with the proprietary software of the tensiometer. The Laplace equation was used for the estimation of the contact angle for both investigated liquids. The series of contact angles observed at 3 s after dispensing were averaged to reduce the scatter of results. However, the range (minimum to maximum) of the observed contact angles was also recorded for further analysis. The surface free energy was computed following OWRK/Fowkes method [30]. The total surface free energy ($\gamma^{tot}$), as well as its polar ($\gamma^{p}$) and disperse ($\gamma^{d}$) components, were determined for all investigated samples.

### 3. Results

The appearance alteration of investigated samples after natural weathering conducted at diverse exposure times is shown in Figure 1. The colour changes are mainly observed for uncoated samples (#A), appearing lighter after three months of exposure. The colour is maintained for the three following months. However, a presence of mould is noticed at month 9 (December). The mould area gradually increased with the subsequent weathering duration. The overall tonality of uncoated samples at the end of the natural weathering test (month 15) became grey. It should be mentioned that in parallel to the above-mentioned colour alterations, samples #A exhibit minor disintegration of surface, evidenced by raised fibres and small cracks. All coated samples performed well, considering the overall appearance alteration as evaluated by the visual assessment. No sign of deterioration was observed in samples #D and #G that were coated with a white and non-transparent film. Similarly, dark brown samples #B and #H did not noticeably change. Conversely, samples #E and #F were coated with a product of original yellow tonality. The colour of coated sample #E become lighter during the initial weathering period to maintain the same appearance afterwards. The colour of coating #C appeared to be constant along the whole testing period.

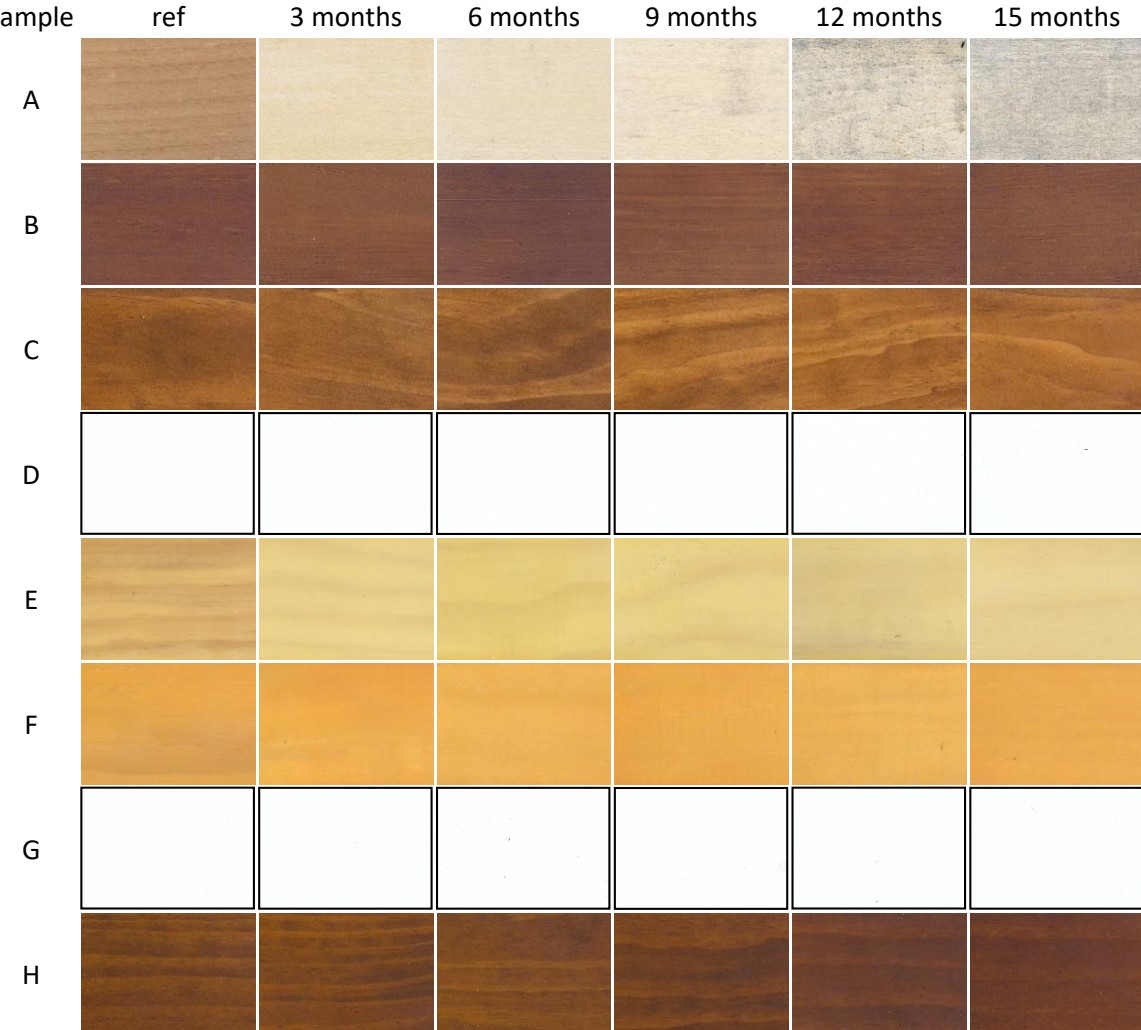

**Figure 1.** Macro colour images of acetylated wood samples exposed to natural weathering.

Figure 2 present colourimetry changes of acetylated wood samples exposed to natural weathering. In analogy to the visual observations, *CIE L\**, *a\** and *b\** are nearly constant for the majority of coated samples along the entire period of the test. It was noticed particularly for samples #D, #G and #H, which are considered here as the most colour-stable. These contain white opaque coatings (provided by producers 1 and 3) as well as a transparent glossy brown coating (producer 3). Slightly higher fluctuations were observed for samples #B and #F. Similar trends were noticed for coated samples #C and #E. *CIE L\** and *CIE b\** values slightly increased during the weathering period in the case of sample #C, while *CIE a\** was relatively constant. A similar tendency was observed in sample #E, where *CIE L\** and *CIE a\** values gradually decreased. *CIE b\** slowly decreased after the initial increase observed at month 3, to reach almost initial values afterwards. The highest changes of colour indicators were noticed for uncoated samples (#A). The apparent lightness (*CIE L\**) noticed for the acetylated radiata pine was relatively stable, with only a slight and steady rise at the initial phase of the weathering test. *CIE a\** gradually decreased for all uncoated samples, reaching more stable values from month 6 of the exposure. Values of *CIE b\** progressively dropped after a slight gradual increase at the beginning of the weathering test.

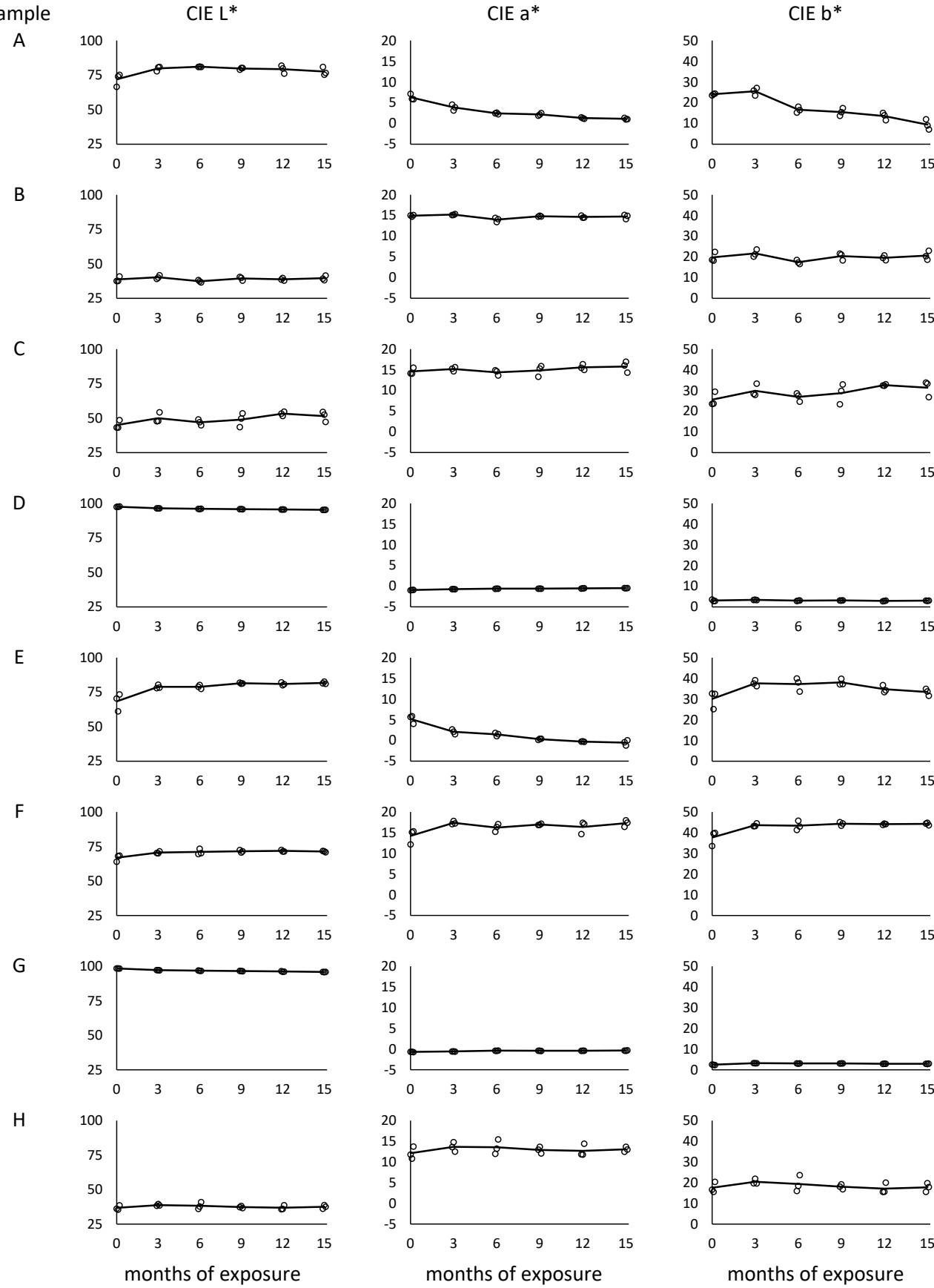

**Figure 2.** Colour changes of acetylated wood samples exposed to natural weathering. A–H: sample type.

Gloss measured along and perpendicular to the fibres (Figure 3) was, in a majority of the investigated cases, very similar. An exception was sample #F where gloss values measured along the fibres were approximately 10 units higher than perpendicular. For coated sample #B, as well as for uncoated samples #A values were rather constant along the whole weathering period. A slight decrease was observed for coated samples #C, #D and #H, contrary to #E, where gloss value increased for about 10 units during the 15 months. Samples #F and #G showed some fluctuations during the weathering period, even if the gloss at the end of the test was close to the initial value. The gloss mainly depended on the performance of the coating and no interaction with the acetylated substrate was evidenced.

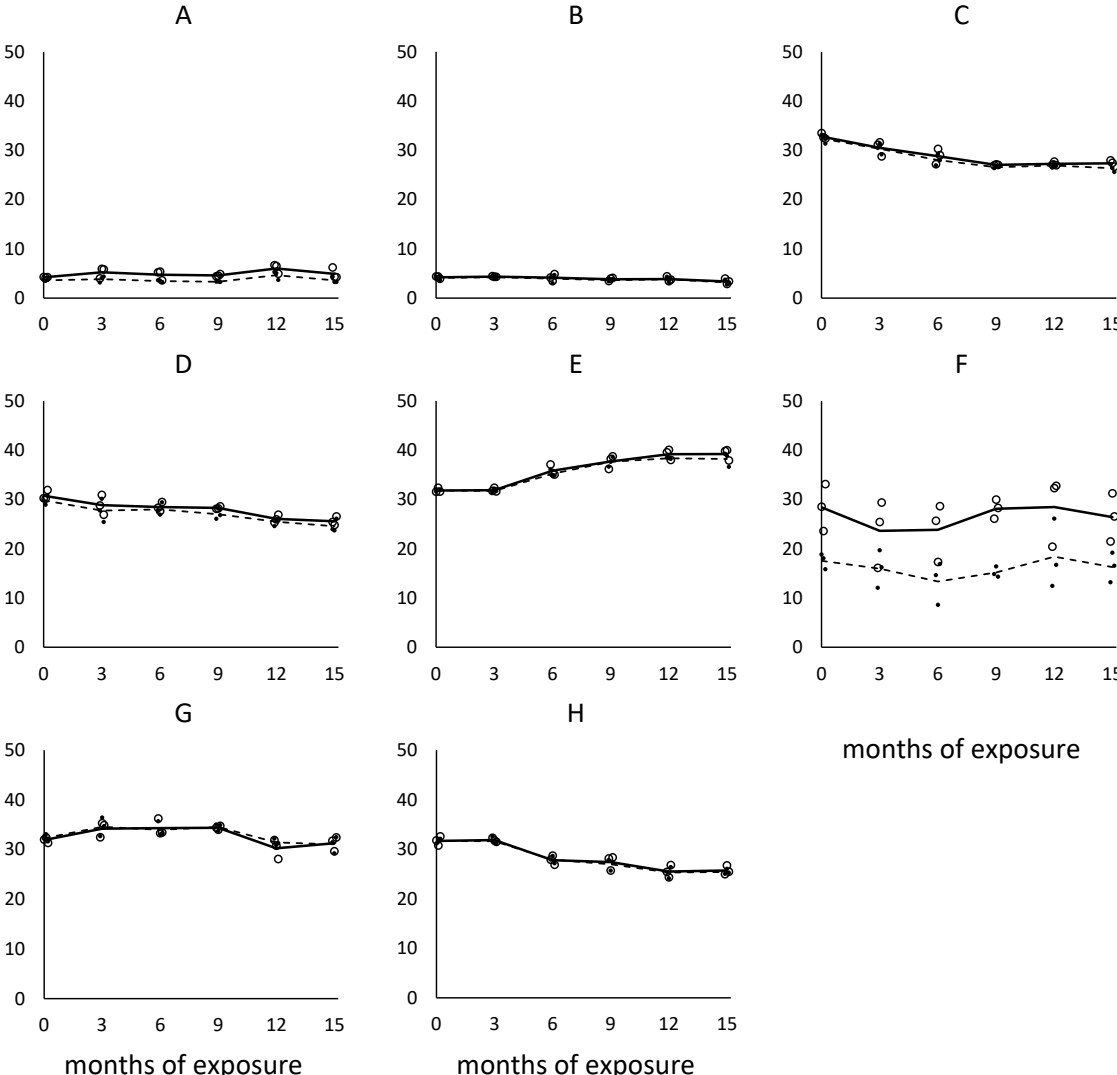

**Figure 3.** Gloss changes of acetylated wood samples exposed to natural weathering (note: solid line–gloss along fibres, dash line–gloss perpendicular to fibres). A–H: sample type.

The surface roughness of the coated wood samples (#B–#H) was not measurably altered during the natural weathering test period. However, an apparent increase in the surface roughness was noticed in uncoated specimens. The result of surface topography analysis performed for samples #A is presented in Figure 4. The 3D topography map aligned with the colour image allows identification of the large tracheids as a major surface irregularity source. However, clear progress of the surface erosion is evident in the analysis of the surface profile outline. It is confirmed as steady progress of the surface roughness parameters. Three-dimensional (3D) areal surface texture assessment is



a favourable approach to characterise surfaces of heterogenic and anisotropic materials, such as wood. Arithmetical mean surface height ($Sa$) is the basic irregularity quantifier corresponding to the arithmetical mean height of the roughness profile ($Ra$) traditionally determined from the two-dimensional surface roughness profiles. Values of $Sa$ remained relatively constant for acetylated radiata pine. Skewness ($Ssk$) represents the degree of the roughness distribution bias or asperity. Negative skewness observed for all measured samples indicates a deviation to the higher side of the topography map that is typical for porous materials, such as wood. The steady decrease of skewness indicates a shift of the top material ratio toward a central distribution that can be associated with the loss of fibres and general progress of the uncoated wood surface erosion. Kurtosis ($Skt$) is a measure of the sharpness of the topography histogram profile. Values of $Skt > 3$ correspond to leptokurtic distribution with tails fatter than in normal/Gaussian curves. The spiked nature of the surface irregularity histogram is a consequence of the flat samples' surfaces generated by the planning operation. The trend of $Skt$ alterations indicates that the curve tended toward platykurtic distribution and a more balanced contribution of the surface peaks and valleys.

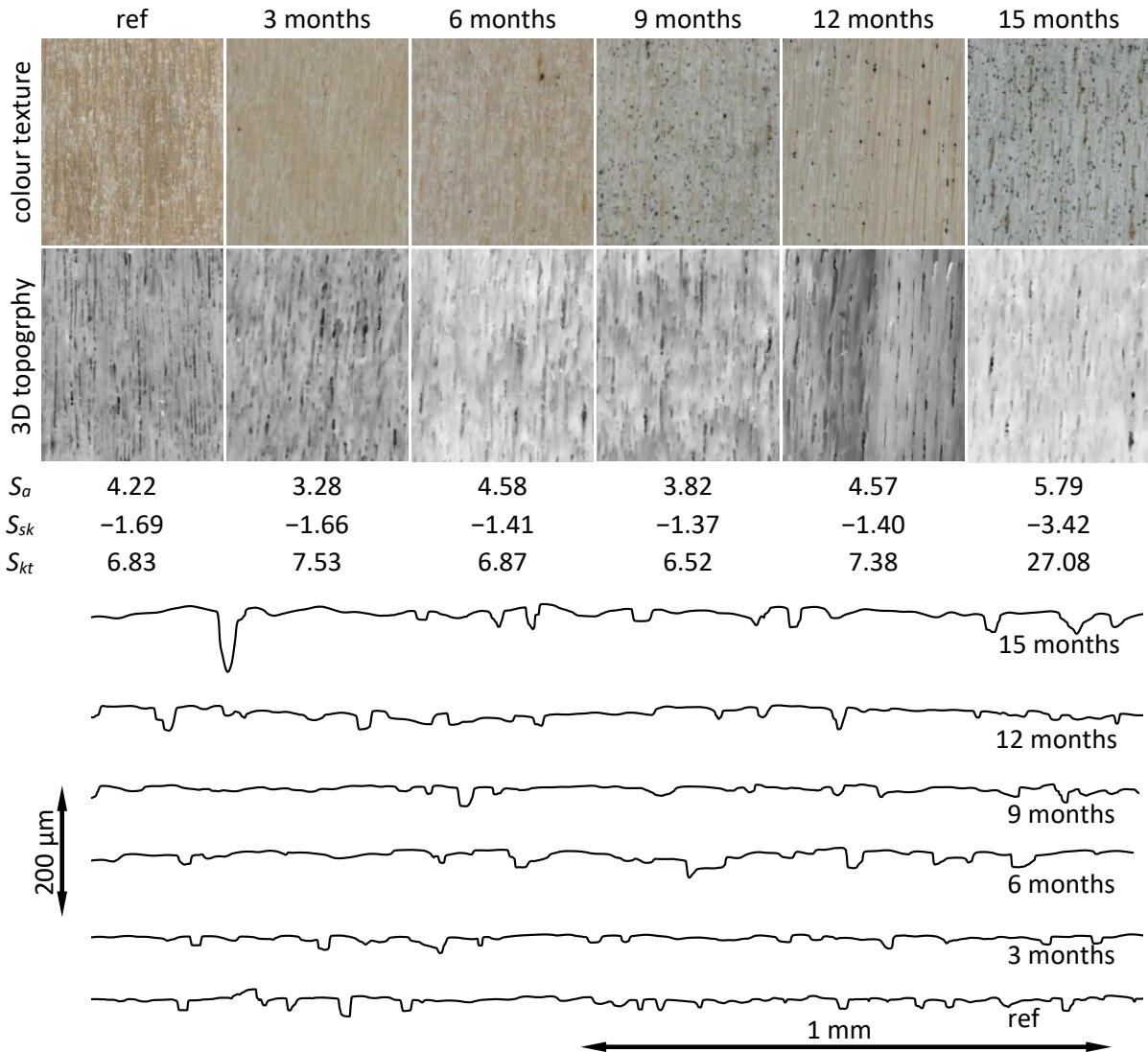

**Figure 4.** 3D surface topography map, typical surface profiles and high magnification images of the acetylated radiata pine wood (sample #A) exposed to natural weathering for a period of 15 months.

An interesting observation is related to the dark spots present on the high magnification colour images. These correspond to the surface contamination and presence of mould

spores. Such spores gradually appeared with the progress of the weathering test that was also perceived on the macro-scale images presented in Figure 1. An evident increase of the infestation kinetics was noticed for all samples exposed for one year or more.

Contact angle (θ) is a quantitative measure of the wetting ability of a solid surface by a liquid. It is defined geometrically as the angle formed by a liquid at the three-phase boundary where the liquid, gas and solid intersect. Results of the dynamic contact angle measurement when wetting the surface of experimental samples with water and formamide are presented in Figure 5. Low contact angle values indicate that the liquid easily spreads over the assessed surface. Conversely, high contact angle implies poor spreading and physical affinity. The high value of θ observed for most of the coated wood samples indicates low wettability by both tested liquids. In all cases, the contact angle measured with water was higher than that with formamide. No clear pattern of the θ changes can be observed on the coated wood samples, even if an initial increase can be noticed in samples #C, #D, #G and #H. A change of the water contact angle at the initial period of the weathering test was observed on the uncoated wood samples (#A). In fact, the surface was so easily wetted by both water and formamide that sessile drops disappeared after a few seconds of the test.

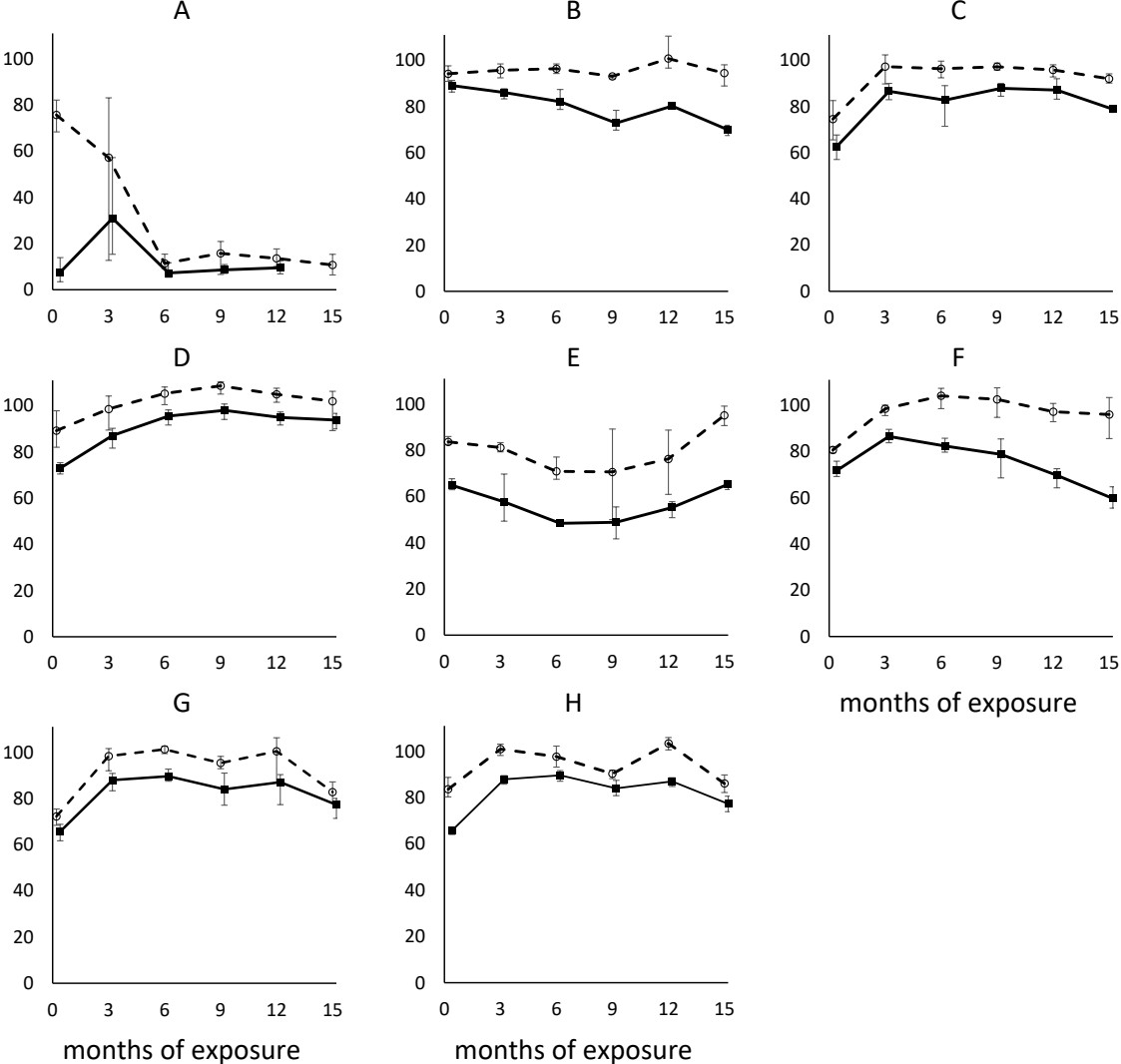

**Figure 5.** Contact angle measured after 3 s of wetting with water (white circle) and formamide (black square). A–H: sample type.

A direct implication of the contact angle measurement is the possibility of the surface free energy (SFE) estimate. SFE was determined following OWRK/Fowkes energy theory that is a recommended method for the coated surfaces. Results of the SFE determined for each measured sample are presented in Figure 6, including total surface free energy ($\gamma^{tot}$) that contains polar ($\gamma^p$) and disperse ($\gamma^d$) components. Values of the SFE were highest for the uncoated wood samples and oscillated around 70 mJ·m$^{-2}$. Even with that, the disperse component dominated in samples #A and #F. An increase of the SFE with the progress of weathering was noticed for coated sample #B and #F. Conversely, the SFE value decreased along with the weathering progress for coated samples #C, #D and #G, while it fluctuated in the case of #E and #H. An increased dispersive part of the modified substrate (acetylated radiata pine wood) and a lower surface free energy after coating application, positively contributed to a stable performance of the coated samples after weathering.

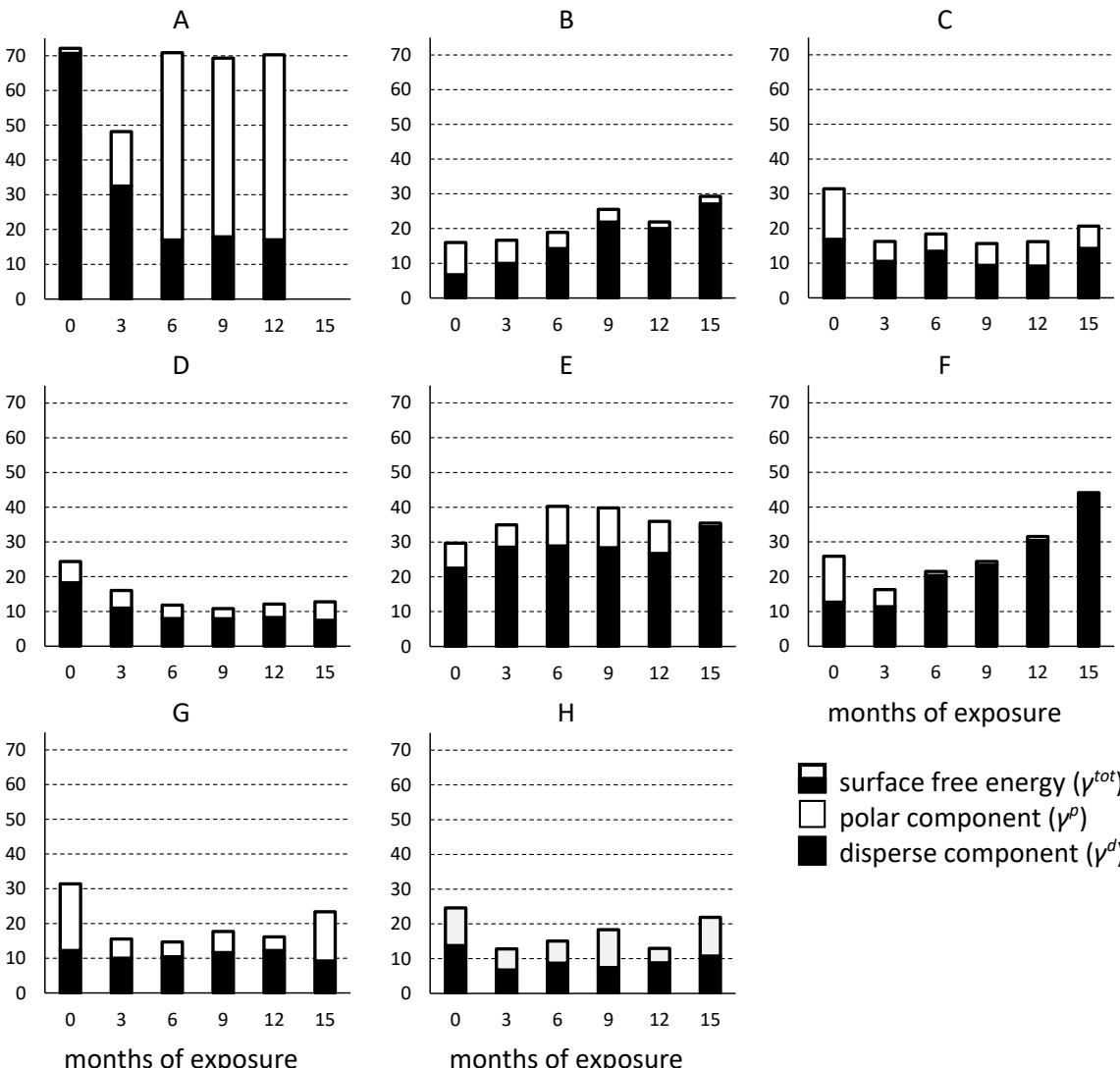

**Figure 6.** Surface free energy ($\gamma^{tot}$) including polar ($\gamma^p$) and disperse ($\gamma^d$) components. A–H: sample type.

## 4. Discussion

All evaluated hybrid modified systems revealed a satisfactory performance during natural weathering tests conducted within this study. Four of seven coatings were transparent, one semi-transparent and two opaques. Transparent coatings are frequently considered attractive solutions for customers since they maintain the natural texture of wood. Though, from the practical implementation perspective, such coating solutions are



not recommended unless regular maintenance is accepted and regularly carried out [31]. Previous studies demonstrated that clear (transparent) coatings failed to give satisfactory performance after just eight months of exposure [21,32]. A detailed overview of the clear coatings' performance applied on the wood surface was provided by Estrada [33]. It was stated that unacceptable appearance is commenced in a relatively short period of 12 to 24 months, depending on the climatic conditions. The main disappointing issue disqualifying clear coatings is associated with the unsatisfactory appeal and necessity for excessive maintenance related to the removal of the old coating layers before re-coating. Clear coatings contain binders that are usually susceptible to photodegradation and/or hydrolysis. That leads to the failure not only to the coating layer but also to the degradation of the underlying wood. It results in discolouration (yellowing) of the surface outlook, loss of glossiness and alteration of the mechanical and physical properties of the coating film [21]. An effective method for improvement of the coating performance is dimension stabilization of the substrate material. The swelling and shrinkage of wood due to changes in its moisture content leads to the generation of checks and cracks within the coating film. Several methods successfully implemented to stabilize the wood and improve the performance of clear coatings were identified and tested experimentally [16,21]. Bulking the wood cell wall with polyethylene glycol or pre-treatment with water repellents as well as various wood modification and impregnation treatments are the best performing solutions. The chemical modification of wood by means of acetylation researched in this study is an example of such an innovative technological approach. It leads to the drastic reduction of native wood hygroscopicity and, consequently, minimizing its shrinkage/swelling. Wood species used as a substrate, type of applied coating, architectural detailing and exposure configuration are known to determine the mould growth on the weathered surfaces [34,35]. No apparent mould presence was noticed in this experiment for all coating systems investigated. Though, evident mould growth was detected after nine months of exposure on the surfaces of uncoated wooden specimens. A similar natural weathering testing campaign with acetylated radiata pine cladding samples was performed in Rotorua (New Zealand) [36]. The first sign of mould presence, as well as slight surface discoloration (lightening), were reported there after only eight weeks of exposure. Conversely, lightening of the acetylated wood surface is declared in the experiment presented here as occurring after the initial three months of exposure. It is related to the methodological approach adopted. Investigated samples were assessed here with scientific instruments after the weathering test (post–factum) by using specimens collected at the three-month interval. Nevertheless, high-resolution images of the same weathered materials exposed on the stand were additionally taken with the frequency of one month [37]. The following image analysis allowed the determination of the red/green/blue (RGB), as well as hue/saturation/luminosity (HSL) colour descriptor variations. It was found that RGB and HSL paraments measured on uncoated acetylated wood samples revealed noticeable alterations after just one month of exposure. Similar research on the natural weathering performance of five wood species coated with 12 diverse formulations was conducted in Oregon (US) for a period of 18 months [38]. It was reported that most of the tested coatings lost their protective effects within the first year of exposure due to combinations of diverse biological and abiotic factors. No single coating system was identified as a superb protective solution for the untreated wood substrate.

Changes in the surface roughness along the weathering progress are associated with the vulnerability to accumulate dirt, moisture and pollution as well as enhancing the spore attachment and germination [14]. Likewise, higher abrasion resistance and improved ability to shed environmental residues deposited on the coating are limiting the ability of fungi to grow on and into the coating [39]. The roughness of hybrid-modified wood surfaces investigated in this research was constant along with the exposure duration. It resulted in negligible appearance changes and minimal dirt/mould presence observed on the coated surface. However, an increase in the surface roughness was observed in uncoated wood specimens. Two from three parameters describing the surface topography

(*Ssk* and *Sku*) changed during weathering progress. It was associated with the removal of the single fibres, leaching of photodegradation components and general erosion of the wood surface. Similar observations were derived in other reports where the roughness of uncoated samples gradually increased with the progress of the weathering process [1].

The change of the surface roughness directly affects the physical characteristics of the light reflected from the surface. It is revealed as gloss changes associated with the abrasion of wood surfaces and the accompanying erosion processes [40,41]. Loss of gloss is an indicator of degradation at its early stage. It is caused by either non-chemical changes (e.g., cracking, checking) or by chemical changes occurring in the topmost portion of the coating layer [42,43]. Therefore, observation of the gloss changes is highly useful for assessing and evaluating coating systems. It was evidenced by Pánek et al. that gloss change is more sensitive to coating degradation than total colour difference measured on the same surface set [44]. Oberhofnerová et al. observed decreased gloss values during natural and artificial weathering of eight transparent and pigmented coating systems [45]. The gloss variation recorded in the hybrid-modified wood samples evaluated in this experiment were minor. It indicates the high stability of the coating systems applied and their satisfactory performance.

The contact angle ($\theta$) measurement implies the wettability by water (or other liquids) of the exposed material surface. The extent of $\theta$ changes is an important indicator of weathering progress [46]. The kinetic and trends of contact angle changes observed in this research for formamide were similar to those of water. Though, the nominal values were higher when wetting the wood surface with water. It is explained by the strong hydrogen bonding of formamide, which reduces the interfacial free energy at the liquid-solid interface through the acid-base interactions [47]. Surface free energy can be calculated by using the $\theta$ data when combing at least two wetting liquids of diverse physical characteristics. This parameter can be utilized to better understand the interactions between solid and liquid. In general, the weathering process of unprotected wood increases wettability. Leaching of the extractives from the surface of weathered wood reduces water repellence, while the degradation of lignin results in a more hydrophilic surface [48]. Different performance trends regarding wettability were reported in relation to the coating transparency and composition [45]. Even if applying the hydrophobic layers significantly increase the hydrophobicity of natural wood surfaces, it does not necessarily indicate long-term functionality during the prospective service life [49].

Bulk and surface modification processes highly affect the suitability and functionality of wood-derived components used in construction of the building façade. Conventional wood modification systems use chemical treatments, impregnation or thermal modification, resulting in various intrinsic properties achieved. Innovative modification approaches focus on adding some extra functionalities, such as UV stabilisation, fire retardancy or enhanced suitability for paints and coatings [50]. Although each modification process improves certain material properties on its own, the positive effect can be multiplied by merging two or more modification solutions. An example of successfully combined modifications is the surface coating of acetylated or thermally treated wood. The synergic effect of the reduced shrinkage/swelling of the bulk substrate and water protecting coating substantially reduces stresses of the coating film, thus preventing it from cracking [51,52] and provides UV-VIS light protection [53]. Such a "hybrid process" of wood modification seems to be an optimal solution frequently implemented by biomaterials producers. Superior service life performance with extended maintenance-free periods is reflected in environmental impact analysis. A recent review of Hill et al. clearly demonstrates a positive impact on the global warming potential (GWM) of wood modification combined with coatings [54]. It is anticipated that a lifetime of unmodified wooden façade is 20 years, assuming coated surface maintenance applied every five years. Conversely, a lifetime of 60 years is assumed for the modified wood solution with the renovation of the coating applied every 10 years. The potential environmental benefit to be realized when utilizing modified wood is evident, especially when considering an extended time between re-coating [54]. It is expected that

the maintenance frequency can be even lower than once per 10 years when implementing the hybrid modification approach [55]. Such an alternative solution was investigated here, where appropriately selected coating systems were applied on the acetylated wooden elements. Consequently, the façade surface remains intact for a longer period by preserving its original and attractive appearance.

## 5. Conclusions

Hybrid modification includes processes where the positive effect of a single treatment can be multiplied by merging with additional modifications. All coated samples performed satisfactorily, considering both, the overall appearance evaluated by the visual assessment as well as deterioration determined with sensors. No sign of degradation was observed in samples #D and #G that were coated with a white and non-transparent film. Dark brown samples #B, #C and #H did not change noticeably. Samples #E and #F were coated with a product of original yellow tonality. The colour of sample #E become lighter during the initial weathering period to maintain the same appearance afterwards. Results of visual evaluations correspond to tendencies observed in *CIE L\**, *a\** and *b\** alterations. Gloss values measured along and perpendicular to fibres were relatively constant. Only a slight decrease was observed for coated samples #C, #D and #H. On the contrary, the gloss value of #E increased for about 10 units during the 15 months exposure. The high values of $\theta$ observed for coated samples indicate low wettability, in contrary to uncoated wood (#A). Surface free energy was highest for the uncoated wood samples and corresponded to ~70 mJ·m$^{-2}$. It varied between 15 and 40 mJ·m$^{-2}$ for coated samples. An increased disperse component $\gamma^d$ combined with a lower surface free energy after coating application resulted in a stable performance of the coated samples during weathering.

It was demonstrated that the hybrid solution of merging wood acetylation process followed by the surface coating, as investigated here, amplified the advantages of both treatments when applied separately. Acetylated wood samples without protective coating altered their appearance noticeably after a relatively short period of exposure to the natural weathering. It was evidenced as a considerable change of the colour, decrease of gloss and increase of roughness. Even if possessing different colours and textures, all coated samples preserved their functionality as a cladding material protecting the building envelope. Limited shrinkage/swelling of the bulk substrate due to chemical treatment substantially reduced stresses within the coating film, therefore no cracks developed in the coating layer. Hybrid processes, compared to solely acetylated wood, assured superior visual performance of the wood surface by preserving its original appearance during whole test duration.

**Author Contributions:** Conceptualization, A.S., E.F.-N., F.P., R.H.D., O.G., N.S., V.P. and J.S.; methodology, A.S., E.F.-N., F.P., R.H.D., O.G., N.S., V.P. and J.S.; software, J.S.; validation, A.S., E.F.-N., F.P., R.H.D., O.G., N.S., V.P. and J.S..; formal analysis, A.S., J.S.; investigation, A.S., E.F.-N., F.P., R.H.D., O.G., N.S., V.P. and J.S.; resources, A.S. and J.S.; data curation, A.S. and J.S.; writing-original draft preparation, A.S.; writing—review and editing, A.S., E.F.-N., F.P., R.H.D., O.G., N.S., V.P. and J.S.; visualization, J.S.; supervision, A.S. and J.S.; project administration, A.S.; funding acquisition, A.S. and J.S. All authors have read and agreed to the published version of the manuscript.

**Funding:** The authors gratefully acknowledge the European Commission for funding the InnoRenew project (grant agreement #739574 under the Horizon2020 Widespread-2-Teaming program), the Republic of Slovenia (investment funding from the Republic of Slovenia and the European Regional Development Fund) and infrastructural ARRS program IO-0035. Part of this work was conducted during project BIO4ever (RBSI14Y7Y4), funded within call SIR by MIUR-Italy; the project Multi-spec (BI-IT/18-20-007), funded by ARRS-Slovenia; Archi-BIO (BI/US-20-054) funded by ARRS-Slovenia, BI-AT/20-21-014 funded by ARRS-Slovenia, J7-9404 (C) funded by ARRS-Slovenia, and CLICK DESIGN, "Delivering fingertip knowledge to enable service life performance specification of wood", (No. 773324) supported under the umbrella of ERA-NET Cofund ForestValue by the Ministry of Education, Science and Sport of the Republic of Slovenia. ForestValue has received funding from the European Union's Horizon 2020 research and innovation programme.

**Institutional Review Board Statement:** Not applicable.

**Informed Consent Statement:** Not applicable.

**Data Availability Statement:** The data presented in this study are available on request from the corresponding author.

**Acknowledgments:** The experimental samples were provided by Accsyss The Netherlands and coated by Teknos/Drywood The Netherlands, ICA Italy, and Renner Italy. Authors would also like to thank Ferry Bongers for valuable comments and discussion.

**Conflicts of Interest:** The authors declare no conflict of interest.

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
