# Peer review of "Hybrid Approach for Wood Modification: Characterization and Evaluation of Weathering Resistance of Coatings on Acetylated Wood"

_coatings, doi:10.3390/coatings11060658_

Round 1

Reviewer 1 Report

Dear Authors,

your study is well designed and clearly described, however it reqires some improvements for better quality in my opinion. Some doubts should be cleared:

L. 37-38 did you measure the aesthetical performance? Or colour and gloss only? There is no clear relationship defined.

L.150 how many samples were in the sets? Which samples they relate?

L.156 8 sample types coded ABCEDEFG

L. 164 "function of exposure time" I expected some correlations, but not calculated.

Fig 2 and 3. What is the meaning of lines? I suppose no meaning. These results should be presented as dot or bar plots. Just like fig 5 or 6.

L. 478 how much are the costs reduced? Did you calculate it? It's not reported, so why cocluded.

In my opinion the conclusion is the weakest part of the manuscript. It look as an abstract. Assumptions and methods of the study should not be re-described. A reader remembers  what was written 2 pages earlier. Please describe and summarize only the new knowledge you odtained basing on your experiments without speculating on cost and aesthetics. Your experiments concerned physical properties only.

Author Response

Reviewer 1

Dear Authors,

your study is well designed and clearly described, however it requires some improvements for better quality in my opinion.

Thank you very much for your kind evaluation of our manuscript and your time dedicated to the review. We have modified the manuscript according to your (and other Reviewer) suggestions. Please find attached the revised version of our work with all the changes in red. There are also some responses to your letter as described below:

Some doubts should be cleared:

  1. 37-38 did you measure the aesthetical performance? Or colour and gloss only? There is no clear relationship defined.

Indeed, it is not clearly stated in the manuscript that we systematically assessed changes of the sample colour, gloss, wettability, roughness, and surface free energy. The selection of these characteristics based on the fact they correspond to material properties directly influencing aesthetical impressions. However, the quantification of the “aesthetical performance” as such was not reported in this manuscript. Following the comment of the Reviewer, we have revised the wording in the abstract replacing “aesthetical” with “visual”.

L.150 how many samples were in the sets? Which samples they relate?

The original materials and methods chapter was perhaps not clear enough. We tested two 8 types of samples representing different coating scenario. Each set included 18 samples corresponding to three replicas that were collected from the stand at pre-defined periods. Therefore, 144 wooden pieces were investigated in total (8 coating scenarios x 6 weathering periods x 3 replicas). One of the coating scenarios was uncoated wood. We do hope that provided augment of the chapter clarify all uncertainties.

L.156 8 sample types coded ABCEDEFG

The text has been modified as suggested Seven alternative coating products manufactured by three producers were tested and confronted; 144 (= 8 × 6 × 3) samples were analysed in total, including 8 sample types (#A, #B, #C, #D, #E, #F, #G, #H), 6 natural weathering scenarios (0, 3, 6, 9, 12 and 15 months) and 3 replicas for each testing scenario.”

  1. 164 "function of exposure time" I expected some correlations, but not calculated.

Perhaps the word “function” is not appropriate in this context, the text has been modified therefore following the Reviewer’s suggestion: “The objective of this test was to collect a set of reference data regarding material performance at the different exposure time.”

Fig 2 and 3. What is the meaning of lines? I suppose no meaning. These results should be presented as dot or bar plots. Just like fig 5 or 6.

Lines linking (averaged) points on the chart are intentionally used to better visualize the trend of changes. We do sincerely believe that it is useful for Readers to observe, particularly some very minor differences (such as in gloss, #A, #D or #E). It is especially justified as each graph is in fact a representation of the (discrete) time-related alterations of the selected property.  Perhaps to provide more consistency to the overall manuscript, we do propose to include such trend lines in Figure 5. We do hope that it is the correct interpretation of the Reviewer’s concern.

  1. 478 how much are the costs reduced? Did you calculate it? It's not reported, so why concluded.

The economic aspects of the implementation of the coated wood solutions were not reported in this manuscript. Following the comment, we decided to revise the conclusion chapter to better reflect the presented content.

In my opinion the conclusion is the weakest part of the manuscript. It look as an abstract. Assumptions and methods of the study should not be re-described. A reader remembers  what was written 2 pages earlier. Please describe and summarize only the new knowledge you obtained basing on your experiments without speculating on cost and aesthetics. Your experiments concerned physical properties only.

The conclusions chapter has been extensively rewritten considering the comments of both Reviewers. Please refer to the re-submitted text where all changes as highlighted in “track changes” mode.

Thank you again for all your comments and suggestions. We did our best to properly address these and do believe that the revised text fulfils the Reviewer’s expectations. We do hope that it can be re-considered for publication in the Coatings journal.

With sincerely regards

Jakub Sandak on behalf of authors

Reviewer 2 Report

Thank you for this interesting and well written manuscript.

1. The conclusion mentions a synergistic effect between acetylation and application of a protective surface coating. The experiment did not include any material that wasn't acetylated so it's not possible to assess synergy. The authors provide background information to support this assertion but do not present any evidence to show whether the effect of the coatings was additive or truly synergistic. I agree with the overall rationale but it should be made clear that this was not tested.

2. Other two minor corrections:

Line 89 change alternate to alter

line 295 change planning to planing.

Author Response

Reviewer 2

Thank you for this interesting and well written manuscript.

Thank you very much for the kind evaluation of our manuscript and your time dedicated to the review. We have modified the manuscript according to your (and other Reviewer) suggestions. Please find attached the revised version of our work with all the changes marked. There are also some specific responses to your letter as described below:

  1. The conclusion mentions a synergistic effect between acetylation and application of a protective surface coating. The experiment did not include any material that wasn't acetylated so it's not possible to assess synergy. The authors provide background information to support this assertion but do not present any evidence to show whether the effect of the coatings was additive or truly synergistic. I agree with the overall rationale but it should be made clear that this was not tested.

We do agree with the critic and therefore extensively revised the conclusion chapter. We do believe that all the problematic statements and not-supported hypotheses are removed. The revised chapter bases entirely on the results presented in the proceeding report.

  1. Other two minor corrections:

Line 89 change alternate to alter

The word is corrected, as suggested.

line 295 change planning to planing.

The word is corrected, as suggested.

Thank you again for all your comments and suggestions. We did our best to properly address both Reviewers’ concerns and do believe that the revised text fulfills expectations. We do hope that it can be re-considered for publication in the Coatings journal.

With sincerely regards

Jakub Sandak on behalf of authors